# Corneal Confocal Microscopy Predicts Cardiovascular and Cerebrovascular Events and Demonstrates Greater Peripheral Neuropathy in Patients with Type 1 Diabetes and Foot Ulcers

**DOI:** 10.3390/diagnostics13172793

**Published:** 2023-08-29

**Authors:** Jonathan Z. M. Lim, Jamie Burgess, Cheong Ooi, Maryam Ferdousi, Shazli Azmi, Alise Kalteniece, Matthew Anson, Daniel J. Cuthbertson, Ioannis N. Petropoulos, Rayaz A. Malik, John P. H. Wilding, Uazman Alam

**Affiliations:** 1Department of Cardiovascular & Metabolic Medicine, Institute of Life Course and Medical Sciences, University of Liverpool, Liverpool L69 3BX, UKj.p.h.wilding@liverpool.ac.uk (J.P.H.W.); uazman.alam@liverpool.ac.uk (U.A.); 2Diabetes, Endocrinology, and Metabolism Centre, Manchester Royal Infirmary, Manchester University NHS Foundation Trust, Manchester M13 9WL, UK; 3Department of Medicine, Clinical Sciences Centre, Aintree University Hospital, Longmoor Lane, Liverpool L9 7AL, UK; 4Institute of Cardiovascular Sciences, Cardiac Centre, Faculty of Medical and Human Sciences, University of Manchester and NIHR/Wellcome Trust Clinical Research Facility, Manchester M13 9WL, UK; 5Liverpool Centre for Cardiovascular Science, University of Liverpool and Liverpool University NHS Foundation Trust, Liverpool L69 3BX, UK; 6Department of Medicine, Weill Cornell Medicine-Qatar, Doha P.O. Box 24144, Qatar; inp2002@qatar-med.cornell.edu (I.N.P.);; 7Centre for Biomechanics and Rehabilitation Technologies, Staffordshire University, Stoke-on-Trent ST4 2DF, UK

**Keywords:** diabetic foot ulcer, diabetic peripheral neuropathy, cardiovascular disease, complications of diabetes, corneal confocal microscopy, type 1 diabetes mellitus

## Abstract

Objective: In this study, we evaluate small and large nerve fibre pathology in relation to diabetic foot ulceration (DFU) and incident cardiovascular and cerebrovascular events in type 1 diabetes (T1D). Methods: A prospective observational study was conducted on people with T1D without diabetic peripheral neuropathy (DPN) (*n* = 25), T1D with DPN (*n* = 28), T1D with DFU (*n* = 25) and 32 healthy volunteers. ROC analysis of parameters was conducted to diagnose DPN and DFU, and multivariate Cox regression analysis was performed to evaluate the predictive ability of corneal nerves for cardiac and cerebrovascular events over 3 years. Results: Corneal nerve fibre length (CNFL), fibre density (CNFD) and branch density (CNBD) were lower in T1D-DPN and T1D-DFU vs. T1D (all *p* < 0.001). In ROC analysis, CNFD (sensitivity 88%, specificity 87%; AUC 0.93; *p* < 0.001; optimal cut-off 7.35 no/mm^2^) and CNFL (sensitivity 76%, specificity 77%; AUC 0.90; *p* < 0.001; optimal cut-off 7.01 mm/mm^2^) had good ability to differentiate T1D with and without DFU. Incident cardiovascular events (*p* < 0.001) and cerebrovascular events (*p* < 0.001) were significantly higher in T1D-DPN and T1D-DFU. Corneal nerve loss, specifically CNFD predicted incident cardiovascular (HR 1.67, 95% CI 1.12 to 2.50, *p* = 0.01) and cerebrovascular (HR 1.55, 95% CI 1.06 to 2.26, *p* = 0.02) events. Conclusions: Our study provides threshold values for corneal nerve fibre metrics for neuropathic foot at risk of DFU and further demonstrates that lower CNFD predicts incident cardiovascular and cerebrovascular events in T1D.

## 1. Introduction

Globally, 537 million adults are living with diabetes, which is a prevalence of 10.5% with a projected rise to 12.2% by 2045 [1]. Diabetic foot ulcers (DFUs) are a major cause of morbidity, mortality and health care expenditure due to hospitalization. The incidence of cardiovascular events is increased in people with DFU, especially those with non-healing or recurrent ulceration [2], with up to a two-fold increase in mortality [3]. 

Small nerve fibre damage precedes large nerve fibre damage in diabetic peripheral neuropathy (DPN) [4]. However, there remains a paucity of data on the natural history and extent of small nerve fibre damage in relation to DFU and there is a lack of consensus as to which measure of DPN best identifies individuals at risk of DFU. Several tests can detect DPN, including vibration perception threshold (VPT), thermal thresholds and nerve conduction studies. Whilst VPT detects DPN and predicts DFU [5], thermal thresholds poorly differentiate patients with and without DFU [6]. The most utilized bedside tests are the 10 g monofilament and 128 Hz tuning fork, both of which identify end-stage neuropathy and those at high risk of DFU. Corneal confocal microscopy (CCM) is a non-invasive ophthalmic test which quantifies small nerve fibre degeneration with reliable and accurate diagnostic ability for DPN [7,8,9]. Previous longitudinal studies have demonstrated the association between corneal nerve fibre loss and progressive large fibre dysfunction [10], suggesting that CCM may predict the development of DFU [11]. Hyperglycaemia is a well-established independent risk factor for cardiovascular disease. However, this risk may be mitigated in people with type 1 diabetes (T1D) with good glycaemic control [12]. Whilst observational studies suggest that a history of DFU increases the risk of cardiovascular disease [13], there is a paucity of evidence on the association between cardiovascular disease and neuropathy. Specifically, there is a lack of high-quality studies showing that CCM-based quantification of small nerve fibre damage is associated with cardiovascular disease.

In this paper, we aimed to determine the diagnostic utility of CCM for DFU and its predictive ability for incident cardiovascular and cerebrovascular events and mortality in patients with T1D.

## 2. Materials and Methods

### 2.1. Study Subjects

This was a prospective observational study of people with T1D attending diabetes outpatient clinics in secondary care. The study was approved by the Preston Research Ethics Committee (REC 18/NW/0532) and sponsored by the University of Liverpool, UK. Written informed consent was obtained according to the Declaration of Helsinki. Participants underwent assessment of the neuropathy symptom profile, neurological examination and nerve conduction studies to determine the presence of neuropathy and active foot ulceration alongside age- and sex-matched healthy volunteers without diabetes. Peripheral neuropathy was identified according to the Toronto criteria [14], and participants in the DFU group had at least one confirmed active diabetic foot ulcer. Participants with other causes of neuropathy (except diabetes), history of neurological disease or previous ocular trauma or ocular surgery were excluded. 

### 2.2. Clinical Assessments

All participants completed a study questionnaire including detailed past medical history and current medications. Participants underwent anthropometric (height, weight and body mass index), and standard clinical care biochemistry as part of standard practice; glycated haemoglobin (HbA_1c_), total cholesterol, HDL cholesterol, triglycerides, renal profile, and urine albumin-to-creatinine ratio (ACR) assessments. 

### 2.3. Assessment of Micro- and Macrovascular Complications 

Screening of both the primary and secondary care health records was undertaken. Retinopathy and maculopathy were identified from the Diabetic Retinopathy National Screening records. Nephropathy was defined according to the estimated glomerular filtration rate (eGFR) and urine albumin-creatinine ratio. Cardiovascular (CV) events were defined based on a history of physician-diagnosed myocardial infarction, angina, coronary artery disease, percutaneous coronary angiography, or coronary artery bypass grafting. Cerebrovascular accidents (CVA) were defined by any new occurrence of ischaemic or haemorrhagic stroke. Mortality data were recorded from the medical records. Macrovascular outcomes were analysed over 3.0 ± 0.6 years of follow-up in all the participants with T1D. 

### 2.4. Assessment of Neuropathy

Peripheral neuropathy was defined according to the Toronto criteria [14] by the presence of an abnormal nerve conduction study and symptom/symptoms and/or sign/signs of peripheral neuropathy. All participants underwent evaluation of neurologic symptoms according to the neuropathy symptom profile (NSP), and the McGill visual analogue score (VAS) was used to assess the severity of painful neuropathy. Neurologic deficits were assessed using the modified neuropathy disability score (NDS), which includes the evaluation of vibration, pin prick, temperature perception and ankle reflexes. Quantitative sensory testing (QST) was performed to determine the cold threshold (CT) (Aδ fibres), warm threshold (WT) (C fibres), cold (CIP) and warm-induced pain (WIP) thresholds using the method of limits with the MEDOC TSA II (Medoc, Ramat Yishai, Israel) on the dorsum of the left foot. Vibration perception threshold (VPT) was measured from an average of three values on the large toe using a neurothesiometer (Horwell, Scientific Laboratory Supplies, Wilford, Nottingham, UK). Participants with a VPT > 15–24 V were considered to have DPN, whilst those with a VPT ≥ 25 V were deemed at high risk for DFU [14]. The sural nerve conduction velocity (SNCV) and sural nerve amplitude (SNAP) were evaluated using the NC-Stat ^®^ DPN Check system (Neurometrix, Waltham, USA). If SNCV and SNAP were unrecordable, the lowest possible recordable value detectable with NC-Stat ^®^ DPN Check system was used (SNCV reading of 28 m/s and SNAP reading of 1.5 μV).

### 2.5. Corneal Confocal Microscopy

All participants underwent CCM for which a laser scanning corneal confocal microscope (Heidelberg Retina Tomograph III; Heidelberg Engineering, Heidelberg, Germany) was used. Several scans of the entire depth of the cornea were recorded using the section mode to acquire and store two-dimensional images with a final resolution of ~2 μm/pixel and an image size of 400 × 400 pixels of sub-basal nerve plexus of the cornea. Eight images (four images from each eye per participant) from the central cornea were selected and examined in a masked and randomised fashion. Automated corneal nerve fibre quantification (ACCMetrics software, version 2.0, University of Manchester, Manchester, UK) was undertaken to derive (1) corneal nerve fibre density (CNFD), number of main nerves/mm^2^ of corneal tissue; (2) corneal nerve branch density (CNBD), number of primary nerve branches/mm^2^); and (3) corneal nerve fibre length (CNFL), length of main nerves and nerve branches (mm/mm^2^) from the eight central cornea nerve images per participant.

### 2.6. Power Calculation 

Corneal nerve fibre density was selected as the primary corneal nerve outcome to assess small nerve fibre structure. With an SD between groups of 9 nerves/mm^2^, we estimated that a minimum of 25 participants for each group would provide an 80% chance to detect a clinically meaningful difference in the CNFD of 5 nerves/mm^2^ and an assumption of a type 1 error (alpha-level) of 0.05. 

### 2.7. Statistical Analysis

Normally distributed data were expressed as mean (standard deviation) (SD). Non-normally distributed data were expressed as median and interquartile range (IQR). Correlations were undertaken using Pearson’s test for normally distributed and Spearman’s rank test for non-normally distributed data. The ANOVA method or a non-parametric counterpart, Kruskal–Wallis, was used to assess differences between groups depending on the normality of the data. Overall, the *p* value was maintained at 0.05 for multiple comparison tests (Bonferroni adjustment or non-parametric counterpart—0.05/4). 

Chi squared tests (r by c Chi squared tests) were undertaken to examine the differences between the microvascular complications at baseline, and incident cardiovascular and cerebrovascular events during the follow-up period. We performed the multivariate Cox regression analysis for neuropathy parameters to evaluate the contribution of each of the categorical and continuous parameter for predicting incident cardiovascular and cerebrovascular events. All selected variables that were predictive, including small and large fibre tests, were entered into the multivariate Cox regression analysis to ascertain the impact on incident cardiovascular and cerebrovascular events. The regression coefficients, hazard ratios, and their corresponding upper and lower 95% confidence interval (CI) were estimated. Receiver operating characteristic (ROC) curve analyses were used to define the Wilcoxon estimate of an area under ROC curve, optimal cut-offs with associated sensitivity and specificity for CCM parameters and small and large fibre tests to identify those with DFU. Statistical analyses were undertaken on SPSS Statistics 25 (IBM Corporation, Armonk, NY, USA).

## 3. Results

### 3.1. Demographics, Metabolic and Anthropometric Assessment (Table 1)

We evaluated 78 people with type 1 diabetes (T1D) without neuropathy, with DPN (T1D-DPN), with active DFU (T1D-DFU) and 32 healthy volunteers without diabetes (Table 1). Participants with T1D-DPN and T1D-DFU were older than healthy volunteers (*p* = 0.012) and had a longer duration of diabetes compared to T1D (*p* < 0.001). HbA_1c_ (*p* < 0.001) was higher in all three groups with diabetes compared to healthy volunteers. The total cholesterol, LDL cholesterol, HDL cholesterol and triglycerides were comparable between all four groups. At baseline, T1D-DPN and T1D-DFU cohorts had a higher prevalence of retinopathy (*p* < 0.001), maculopathy (*p* < 0.001) and nephropathy (*p* < 0.001).

**Table 1 diagnostics-13-02793-t001:** Demographics and metabolic profile of the participants.

	Healthy Volunteers (*n* = 32)	T1D (*n* = 25)	DPN (*n* = 28)	DFU (*n* = 25)	*p* Value T1D vs. DPN	*p* Value DPN vs. DFU	*p* Value All Groups
Demographics							
Age (years)	41.1 (11.3)	43.4 (13.2)	48.3 (7.8)	49.8 (9.2)	0.18	0.33	0.01
Sex (Female) (no.) (%)	17 (53)	11 (44)	14 (50)	4 (16)	0.67	0.01	0.03
Duration of diabetes (years)	0 (0)	16.2 (12.3)	25.5 (11.7)	26.9 (10.9)	0.01	0.67	<0.001
BMI (kg/m^2^)	24.50 (3.8)	27.8 (4.9)	28.9 (5.7)	28.2 (5.7)	0.49	0.33	0.42
Biochemistry							
HbA_1c_ (mmol/mol)	37 (4)	69 (13)	78 (16)	80 (17)	0.03	0.74	<0.001
Cholesterol (mmol/L)	4.7 (0.7)	4.4 (1.0)	4.8 (1.2)	4.9 (0.5)	0.18	0.68	0.22
HDL (mmol/L)	1.4 (0.4)	1.7 (0.4)	1.7 (0.5)	1.5 (0.3)	0.98	0.22	0.21
LDL (mmol/L)	2.6 (0.8)	2.0 (0.8)	2.4 (0.9)	2.2 (0.6)	0.09	0.26	0.05
Triglycerides (mmol/L)	1.4 (0.7)	1.5 (0.8)	1.7 (1.1)	1.4 (0.5)	0.34	0.24	0.61
eGFR (ml/min/1.73 m^2^)	82 (13)	81 (17)	74 (20)	75 (20)	0.17	0.78	0.30
UACR (mg/mmol), median (IQR)	0 (0)	0.7 (1.4)	2.4 (12.1)	23.3 (37.3)	0.06	0.002	<0.001
Microvascular Complications							
Nephropathy (CKD 3–5) (no.) (%)	0 (0)	5 (20)	9 (32)	11 (44)	0.32	0.37	<0.001
Retinopathy (R1–R3) (no.) (%)	0 (0)	7 (28)	18 (64)	18 (72)	0.01	0.49	<0.001
Maculopathy (M1) (no.) (%)	0 (0)	5 (20)	14 (50)	16 (64)	0.02	0.31	<0.001

Table key: Values represented as mean (standard deviation) (SD) unless otherwise stated. T1D—participants with type 1 diabetes; DPN—participants with T1D and diabetic peripheral neuropathy; DFU—participants with T1D and diabetic foot ulceration; BMI—body mass index; eGFR- estimated glomerular filtration rate; UACR – urine albumin creatinine ratio; HbA_1c_—glycated haemoglobin; HDL—high-density lipoprotein; LDL—low-density lipoprotein; T1D—type 1 diabetes; T1D-DPN—type 1 diabetes with diabetic peripheral neuropathy; T1D-DFU—type 1 diabetes with diabetic foot ulcer.

### 3.2. Neuropathy Assessment (Table 2)

Participants with T1D-DPN and T1D-DFU had higher VPT and WT and lower SNCV and SNAP compared to those without neuropathy and healthy volunteers (Table 2). Participants with T1D-DFU and T1D-DPN had higher VPT (*p* < 0.001), lower SNCV (*p* < 0.001), but comparable SNAP (*p* = 0.147). Corneal nerve fibre density (CNFD), corneal nerve fibre length (CNFL) and corneal nerve branch density (CNBD) were lower in T1D-DPN and T1D-DFU compared to T1D (*p* < 0.001) and were lower in T1D-DFU compared to T1D-DPN (Table 2). Corneal nerve images from healthy volunteers, T1D, T1D-DPN and T1D-DFU are presented in Figure 1 and the progressive loss of corneal nerve fibres in each group are displayed in Figure 2a–c.

**Table 2 diagnostics-13-02793-t002:** Neuropathy symptoms, neurological deficits, quantitative sensory testing, neurophysiology and corneal confocal microscopy in all participants.

	Healthy Volunteers (*n* = 32)	T1D (*n* = 25)	DPN (*n* = 28)	DFU (*n* = 25)	*p* Value T1D vs. DPN	*p* Value DPN vs. DFU	*p* Value All Groups
VAS	0 (0)	0.6 (1.2)	5.3 (3.0)	5.8 (2.6)	<0.001	0.452	<0.001
NSP	0 (0)	2.1 (2.6)	11.9 (7.2)	18.2 (5.1)	<0.001	<0.001	<0.001
NDS	0 (0)	0.6 (0.8)	5.2 (2.6)	7.8 (2.3)	<0.001	<0.001	<0.001
VPT (Volt)	6.7 (2.8)	9.4 (2.6)	17.2 (4.3)	24.9 (6.7)	<0.001	<0.001	<0.001
SNCV (m/s)	55.5 (2.3)	50.8 (4.5)	35.4 (3.9)	30.7 (2.6)	<0.001	<0.001	<0.001
SNAP (μV)	13.6 (1.6)	7.5 (3.5)	3.0 (2.0)	2.2 (1.0)	<0.001	0.062	<0.001
CT (°C)	27.6 (2.8)	26.6 (1.9)	17.1 (5.2)	12.8 (2.8)	<0.001	0.001	<0.001
WT (°C)	33.7 (1.3)	37.6 (2.1)	41.2 (1.4)	46.3 (3.5)	<0.001	<0.001	<0.001
CIP (°C)	15.4 (2.7)	16.0 (4.2)	9.3 (4.6)	3.2 (2.8)	<0.001	<0.001	<0.001
WIP (°C)	38.3 (1.1)	41.1 (1.6)	44.1 (1.8)	48.4 (2.0)	<0.001	<0.001	<0.001
CNFL (mm/mm^2^)	21.08 (2.77)	20.35 (2.46)	9.78 (4.58)	5.17 (1.82)	<0.001	<0.001	<0.001
CNFD (no./mm^2^)	25.02 (4.27)	23.06 (6.21)	11.50 (5.08)	6.00 (2.59)	<0.001	<0.001	<0.001
CNBD (no./mm^2^)	26.94 (7.28)	21.53 (7.27)	11.51 (6.84)	5.71 (3.43)	<0.001	<0.001	<0.001

Table key: Values represented as mean (SD) unless otherwise stated. T1D—participants with type 1 diabetes; DPN—participants with T1D and diabetic peripheral neuropathy; DFU—participants with T1D and diabetic foot ulcer; VAS—McGill Visual Analogue Score for pain (out of maximum score of 10); NSP—neuropathy symptom profile (out of maximum score of 38); NDS—Neuropathy Disability Score (out of maximum score of 10); VPT—vibration perception threshold; SNCV—sural nerve conduction velocity; SNAP—sural nerve amplitude; CT—Cold Threshold; WT—warm threshold; CIP—cold-induced pain; WIP—warm-induced pain; CNFL—corneal nerve fibre length; CNFD—corneal nerve fibre density; CNBD—corneal nerve branch density; T1D—type 1 diabetes; DPN—type 1 diabetes with diabetic peripheral neuropathy; DFU—type 1 diabetes with diabetic foot ulcer.

### 3.3. Receiver-Operating Characteristic (ROC) Analysis to Establish Diagnostic Utility of QST, NCV and CCM Parameters for DFU and DPN

To compare the diagnostic ability of the different tests for DFU, ROC analysis was undertaken. ROC analysis showed that CNFL (Sn/Sp: 0.76/0.77; AUC 0.90; *p* < 0.001), CNFD (Sn/Sp: 0.88/0.87; AUC 0.93; *p* < 0.001) and CNBD (Sn/Sp: 0.84/0.74; AUC 0.86; *p* < 0.001) demonstrated good diagnostic ability for DFU (Figure 3a). VPT, CT, WT, SNCV and SNAP each demonstrated good diagnostic ability for DFU (Figure 3b). Diagnostic accuracy for identifying patients with DFU was comparable between small nerve fibre (CT, WT, CNFL, CNFD and CNBD) and large nerve fibre (SNCV, SNAP and VPT) diagnostic tests (Table 3).

The ROC analysis demonstrated that CNFL (Sn/Sp: 0.96/0.92; AUC 0.98; *p* < 0.001), CNFD (Sn/Sp: 0.86/0.80; AUC 0.93; *p* < 0.001) and CNBD (Sn/Sp: 0.75/0.68; AUC 0.83; *p* < 0.001) had good-to-excellent diagnostic ability for DPN (Appendix A). Small nerve fibre diagnostic tests (CT, WT, CNFL and CNFD) had comparable diagnostic accuracy to large nerve fibre diagnostic test (VPT) for identifying patients with DPN. Given that SNCV and SNAP formed a part of the diagnostic criteria for DPN they were excluded from this ROC analysis (Appendix A).

### 3.4. Predictors of Cardiovascular and Cerebrovascular Events in Patients with T1D

Over a 3-year follow-up, the incidence of new cardiovascular events (*p* < 0.001), cerebrovascular events (*p* < 0.001) and lower extremity amputation (*p* < 0.001) were higher in T1D-DPN and T1D-DFU (Table 4). There was no difference in the 3-year mortality rates (*p* = 0.25) between groups. In the Cox-based regression analysis, we evaluated the key parameters and characteristics of T1D participants (without DPN, with DPN and DFU) for developing new incident cardiovascular and cerebrovascular events, respectively (Appendix A). In the multivariate Cox regression analysis, decreased CNFD predicted greater incident cardiovascular (HR 1.67, 95% CI 1.12–2.50, *p* = 0.01) and cerebrovascular (HR 1.55, 95% CI 1.06–2.26, *p* = 0.02) events (Table 5). 

## 4. Discussion

In this study, we report greater small and large nerve fibre damage in T1D patients with DFU, with a clear association between corneal nerve fibre loss and the severity of DPN. We further demonstrate for the first time that corneal nerve loss predicts incident cardiovascular and cerebrovascular events, more so than age, duration of diabetes, dyslipidaemia, and other measures of neuropathy, including nerve conduction studies and VPT. Our data suggest that degeneration of corneal nerve fibres may serve as a surrogate biomarker for detecting the at-risk diabetic foot and higher incidence of cardiovascular and cerebrovascular events, though the mechanistic association remains to be established.

Both large and small nerve fibre degeneration occurs in DPN [4,6], though small fibres are affected earlier than larger fibres [15]. Neuropathy and insensitivity to trauma is key to the development of DFU [11,16]. Indeed, reduced motor nerve conduction velocity has been shown to predict foot ulceration and increased mortality in diabetes [17] and increased VPT has a good predictive value for the development of DFU [14]. Small nerve fibre damage with impaired heat and pain sensation exposes patients to unperceived foot trauma [18] and increased thermal thresholds are associated with an increased likelihood of DFU [19]. Therefore, sensory and autonomic neuropathy independently influence the risk of DFU and amputation [20,21]. However, impaired small fibre-mediated pressure-induced vasodilation, the hyperaemic response to tissue injury [19,22] and deficiency in nerve growth factor and substance P are key mediators of skin breakdown and blunted healing of DFU [21,23,24]. The evaluation of small fibre neuropathy has relied on quantitative sensory testing (QST) and contact heat-evoked potential tests, but they are time consuming and subject to large intra-individual variability. Skin biopsy and assessment of intraepidermal nerve fibre density (IENFD) allows to conduct a direct measure of small fibre pathology in DPN, but it is an invasive technique [25]. We have previously shown that CNFD and IENFD have comparable diagnostic utility for DPN [26]. This study now demonstrates that CNFD, CNBD and CNFL have good diagnostic utility for patients with DFU, consistent with our data showing that a reduction in CNFL preceded the development of DFU [11]. A recent meta-analysis has confirmed that CCM has good diagnostic utility for DPN [9]. Furthermore, lower CNFL predicts 4-year incident DPN [27] and a more rapid CNFL decline is associated with the development of DPN [28] and foot ulceration [11,16]. We now show that CNFL is a superior diagnostic biomarker for DPN (Sn 0.96, Sp 0.92; AUC 0.98; (95% CI) (0.96–1.00); *p* < 0.001), whilst CNFD is superior for DFU (Sn 0.88, Sp 0.87; AUC 0.93; (95% CI) (0.88–0.98); *p* < 0.001). This suggests an initial length dependent process followed by more global proximal nerve fibre loss with more severe DPN, reflecting a similar process in intra-epidermal nerve fibres. 

Inflammation and premature atherosclerosis remain the predominant cause of excess mortality in T1D. Impaired flow-mediated dilation (FMD) and reactive hyperaemia peripheral artery tonometry (RH-PAT) are associated with small fibre denervation and dysfunction. Small fibre deficits are also associated with a higher incidence of CAN [29] with an increased risk of myocardial dysfunction, silent myocardial ischaemia and cardiac arrhythmias [29]. In our previous studies we have shown greater corneal nerve loss in patients with acute ischemic stroke [15], especially those with recurrent stroke [30]. In the Canadian cohort study of people living with T1D for over 50 years, greater coronary artery calcification [31] was associated with large fibre neuropathy and retinopathy, suggesting common inflammatory pathways. The mechanisms underpinning the association between small nerve fibre degeneration and atherosclerotic cardiovascular disease remain unclear. Atherosclerotic plaque inflammation leads to the transmission of impulses via sensory afferent fibres to the medullary and hypothalamic neurones, thus increases sympathetic efferent activity in the vessel wall (artery-brain circuit), resulting in increased lymphocyte and cytokine activity and inflammation [32]. Ganglionectomy in experimental models has been shown to attenuate this neuroimmune activation by the peripheral nervous system and reduce disease progression [32].

Limitations of our study include the potential influence of confounding factors such as the long duration of diabetes, predisposing to increased risk of DFU and cardiovascular disease. We also investigated relatively small numbers of patients with minor disparities in age between the cohorts. Studies to assess the utility of CCM in the prediction of cardiovascular disease and recurrent DFU are warranted. Further studies involving the use of artificial intelligence (AI)-based algorithms and end-to-end classification of CCM images [7] may strengthen the utility of CCM. Recently, machine learning algorithm was used to develop systems based on several discriminative patient parameters, helping identify patients at high risk of cardiovascular events [33]. 

In summary, we show that small nerve fibre damage is prevalent in patients with DFU and present diagnostic cut-offs for CCM to identify the at-risk neuropathic foot. Furthermore, we show that corneal nerve fibre density predicts incident cardiovascular and cerebrovascular events in patients with T1D. 

## Figures and Tables

**Figure 1 diagnostics-13-02793-f001:**
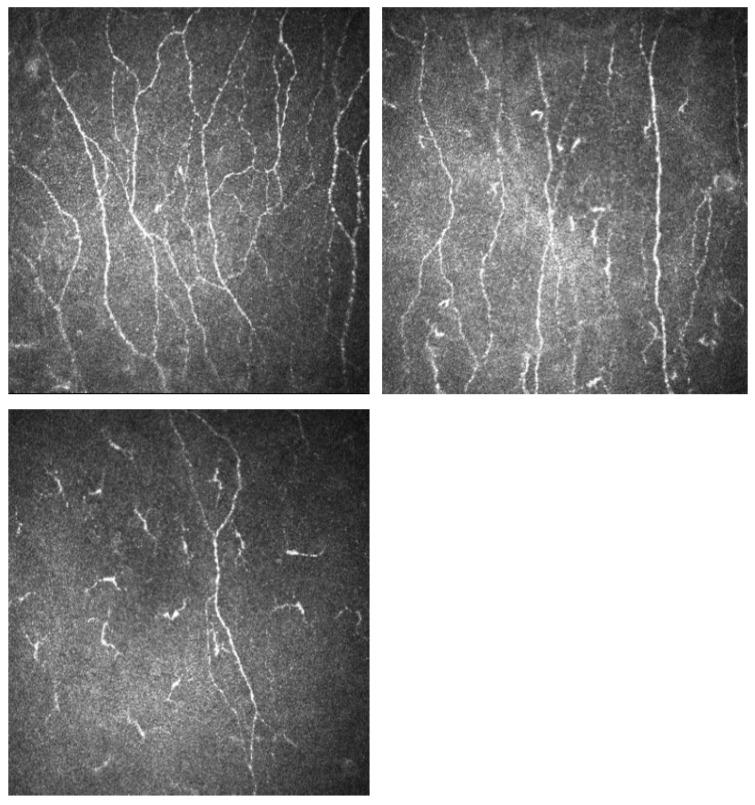
Corneal confocal microscopy images of the sub-basal nerve plexus showing a progressive loss of corneal nerves in a patient with T1D (**top right**), T1D-DPN (bottom left) and T1D-DFU (**bottom right**) compared to a healthy control (**top left**). T1D—type 1 diabetes; T1D-DPN—type 1 diabetes with diabetic peripheral neuropathy; T1D-DFU—type 1 diabetes with diabetic foot ulcer.

**Figure 2 diagnostics-13-02793-f002:**
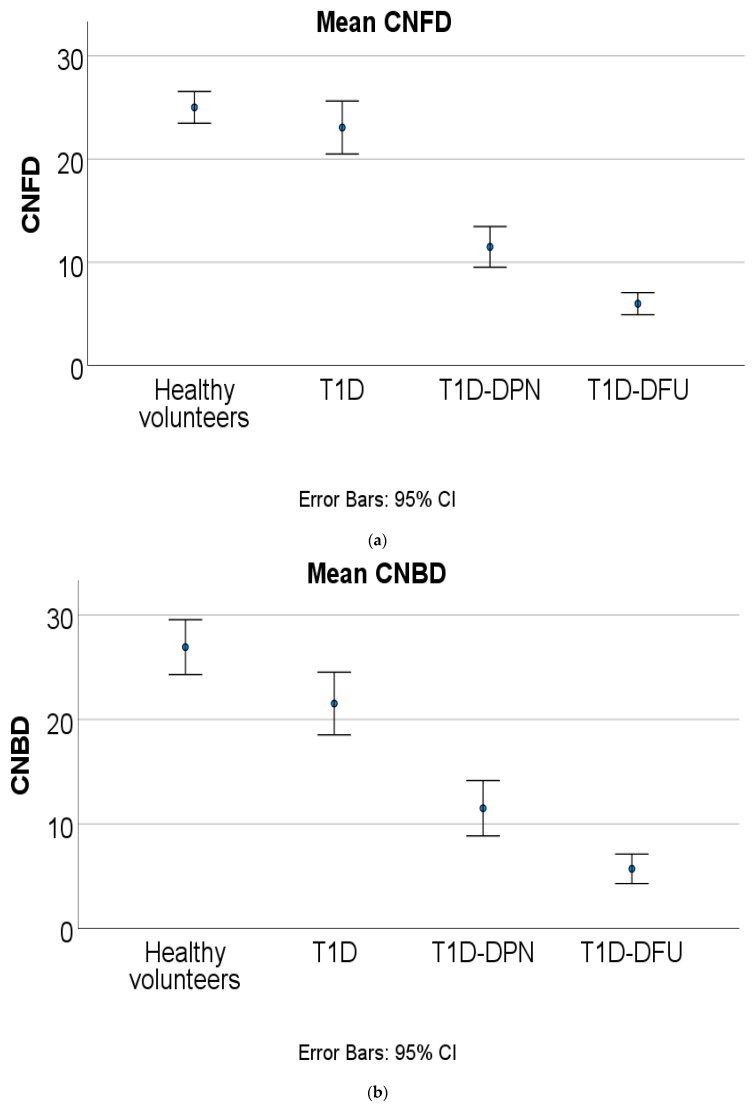
(**a**) Comparison of corneal nerve fibre density (CNFD) between groups. (**b**) Comparison of corneal nerve branch density (CNBD) between groups. (**c**) Comparison of corneal nerve fibre length (CNFL) between groups. T1D—type 1 diabetes; T1D-DPN—type 1 diabetes with diabetic peripheral neuropathy; T1D-DFU—type 1 diabetes with diabetic foot ulcer.

**Figure 3 diagnostics-13-02793-f003:**
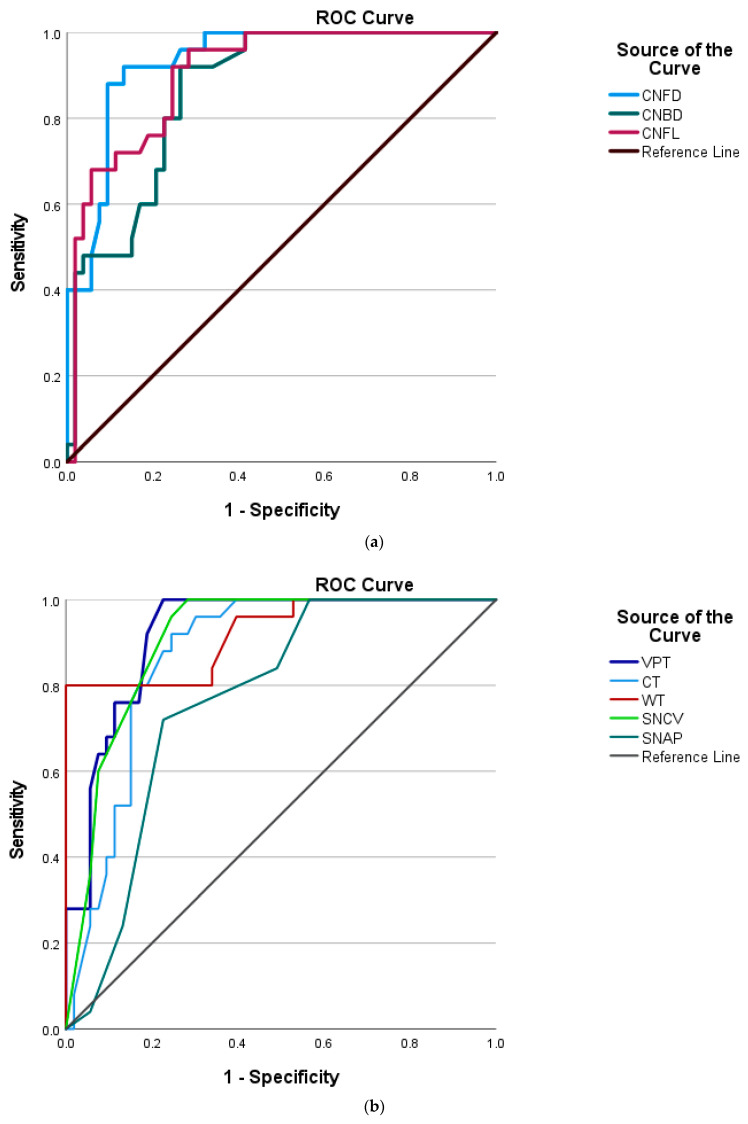
(**a**) ROC curves for CNFD, CNBD and CNFL for T1D-DFU. (**b**) ROC curves for VPT, CT, WT, SNCV and SNAP for T1D-DFU. CNFD—corneal nerve fibre density; CNBD—corneal nerve branch density; CNFL—corneal nerve fibre length; VPT—vibration perception threshold; CT—cold threshold; WT—warm threshold; SNCV—sural nerve conduction velocity; SNAP—sural nerve amplitude.

**Table 3 diagnostics-13-02793-t003:** ROC analysis with AUC, optimal cut off, sensitivity and specificity with 95% confidence interval in T1D with and without DFU.

	Optimal Cut off	Sensitivity	Specificity	AUC (95% CI)	*p* Value
CNFD (no./mm^2^)	7.35	0.88	0.87	0.93 (0.88–0.98)	<0.001
CNBD (no./mm^2^)	7.57	0.84	0.74	0.86 (0.78–0.94)	<0.001
CNFL (mm/mm^2^)	7.01	0.76	0.77	0.90 (0.84–0.97)	<0.001
VPT (Volts)	18.3	0.92	0.81	0.92 (0.86–0.98)	<0.001
CT (°C)	15.8	0.80	0.81	0.87 (0.79–0.95)	<0.001
WT (°C)	42.2	0.80	0.89	0.92 (0.85–0.99)	<0.001
SNCV (m/s)	35.0	0.96	0.76	0.91 (0.84–0.97)	<0.001
SNAP (μV)	2.5	0.72	0.77	0.77 (0.66–0.87)	<0.001

AUC—area under the curve; CNFL—corneal nerve fibre length; CNFD—corneal nerve fibre density; CNBD—corneal nerve branch density; T1D—type 1 diabetes; T1D-DFU—type 1 diabetes and diabetic foot ulcer; VPT—vibration perception threshold; CT—cold threshold; WT—warm threshold; SNCV—sural nerve conduction velocity; SNAP—sural nerve amplitude.

**Table 4 diagnostics-13-02793-t004:** Morbidity and mortality outcomes in participants with T1D.

	T1D (*n* = 25)	T1D-DPN (*n* = 28)	T1D-DFU (*n* = 25)	*p* Value between All Groups
Duration of follow-up (years), mean (SD)	3.0 (0.7)	3.1 (0.6)	3.1 (0.6)	0.48
Lower extremity amputation (no. of events)	0	4	12	<0.001
Cardiovascular events (no. of events)	0	5	12	<0.001
Cerebrovascular events (no. of events)	0	5	11	<0.001
Mortality (no. of cases)	0	1	3	0.25

**Table 5 diagnostics-13-02793-t005:** Associations with incident cardiovascular and cerebrovascular events over 3 years in patients with T1D based on the multivariate Cox regression analysis.

Parameters	HR (95% CI)	*p* Value
Cardiovascular Events		
Model (X^2^ = 34.8, *p* < 0.001)		
Age ^α^	1.07 (0.96–1.19)	0.22
Gender (Male)	3.67 (0.76–17.86)	0.11
Duration of T1D ^α^	1.07 (0.98–1.16)	0.13
SNCV ^α^	1.14 (0.91–1.44)	0.26
WT ^α^	1.11 (0.93–1.33)	0.26
VPT ^α^	1.04 (0.94–1.15)	0.45
CNFD ^β^	1.67 (1.12–2.50)	0.01
Cerebrovascular Events		
Model (X^2^ = 30.1, *p* < 0.001)		
Age ^α^	1.07 (0.97–1.19)	0.20
Gender (Male)	3.95 (0.71–22.22)	0.12
Duration of T1D ^α^	1.03 (0.94–1.12)	0.51
SNCV ^α^	1.12 (0.89–1.40)	0.34
WT ^α^	1.12 (0.93–1.35)	0.24
VPT ^α^	1.04 (0.94–1.14)	0.43
CNFD ^β^	1.55 (1.06–2.26)	0.02

^α^—increased/higher value was associated with increased incidence of cardiovascular and cerebrovascular events. ^β^—decreased/lower value was associated with increased incidence of cardiovascular and cerebrovascular events.

## Data Availability

Summary data supporting this study are included within the article and/or supporting materials. Additional data are available on reasonable request but may warrant data transfer agreements and costs may be incurred.

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
