# Peer review of "Corneal Confocal Microscopy Predicts Cardiovascular and Cerebrovascular Events and Demonstrates Greater Peripheral Neuropathy in Patients with Type 1 Diabetes and Foot Ulcers"

_diagnostics, 2023, doi:10.3390/diagnostics13172793_

Round 1

Reviewer 1 Report

Please find attached the revised paper with some comments on it

Moderate- minor editing of English language required

Author Response

Response to Reviewer 1 Comments

We thank the reviewer to using this opportunity to provide an overview and response to the manuscript.

DPN not previously defined.

Thank you. We have defined the acronym in the abstract.

Statistical analysis. It is a little bit complex the understanding. I think is better explain the statistical analysis by comparison and in paragraphs.

We have made it into separate paragraphs for ease of reading.  The paragraphs use standard statistical terms to describe the analytical methods used.

The table (Table 1) is difficult to read. Please choose the comparison and expressed in groups. For example, one table with T1D and T1D-DFU and other with DPN and DFU.

Thank you for the feedback and response the reviewer has provided for the change in the table layouts. As this study was a direct observational comparison between the four groups, particular focus was on the observed differences between T1D vs T1D-DPN, and DPN vs DFU. We felt that as there were already several tables overall it would not be helpful for understanding of our work to split the tables into further sub-comparisons. We felt the group comparisons and P value at the end was the most straightforward way of displaying the results, so as not to duplicate data into two further tables.

The information in text and the table 2 is the same, is repetitive. Please choose one. I think is better the table.

Thank you. We have removed phrases which were deem repetitive to reduce burden on the readers. Contents and statistical values displayed within the tables are referred to in the text.

The information is duplicated in the text and in the figure and table. The information should be simplified and no duplicated

Thank you. We have simplified the description of results within the text.

The information is duplicated in the text, figures, and table. Please, choose one. It is difficult to understand the results (section 3.3, 3.4)

Thank you. We have simplified the description of results within the text.

Results. Table for comparison and expressed in groups. Choose the comparison.

Discussion. Our results suggest.... I think that is a big affirmation. All the discussion should be in reference of their results.

Thank you. We have altered the wording to reflect the observation and findings from our results only.

Reviewer 2 Report

Authors studied Corneal confocal microscopy in prediction of foot ulcers in patients with type 1 DM. In general, it is a well written study. My comments are as follows:

- I am not sure the sections what is already known about this subject?, What are the new findings?, and How might this impact on clinical practice in the foreseeable future? are necessary. IF not, please remove. 

- Abstraction of the text and keywords are adequate.

- Objectives and rationale of the study are clear in introduction with adequate background data

- Methodology, including statistics, is presented very well.

- Results make sense. It looks like the objectives are met.

- Discussion is also fair but could be better. Since both diabetic neuropathy and diabetic foot ulcers are characterized with increased inflammatory burden, I suggest authors to interpret and discuss the results of the study in the context of inflammation. Moreover, clinical translation of the study results could be emphasized in discussion.

Author Response

Feedback and Response to reviewer 2

Authors studied Corneal confocal microscopy in prediction of foot ulcers in patients with type 1 DM. In general, it is a well written study. My comments are as follows:

- I am not sure the sections what is already known about this subject? What are the new findings? How might this impact on clinical practice in the foreseeable future? are necessary. IF not, please remove. 

Thank you for the reviewer’s comments. We have removed this section as deemed unnecessary for the format of this journal.

- Abstraction of the text and keywords are adequate.

Thank you.

- Objectives and rationale of the study are clear in introduction with adequate background data

Thank you.

- Methodology, including statistics, is presented very well.

Thank you.

- Results make sense. It looks like the objectives are met.

Thank you.

- Discussion is also fair but could be better. Since both diabetic neuropathy and diabetic foot ulcers are characterized with increased inflammatory burden, I suggest authors to interpret and discuss the results of the study in the context of inflammation. Moreover, clinical translation of the study results could be emphasized in discussion.

Interpret and discuss the results of the study in context of inflammation.

Thank you. We have made some modifications and improved the focus on neuro-inflammation in relation to the cardiovascular and cerebrovascular events with regards to nerve fibre inflammation.

Round 2

Reviewer 1 Report

The paper has been improved with the coments and in my opinión is suitable for publication 

The english lenguaje IS good enought